# Comparative Efficacy of Exosomes Derived from Different Mesenchymal Stem Cell Sources in Osteoarthritis Models: An In Vitro and Ex Vivo Analysis

**DOI:** 10.3390/ijms26125447

**Published:** 2025-06-06

**Authors:** Jaishree Sankaranarayanan, Hyung Keun Kim, Ju Yeon Kang, Sree Samanvitha Kuppa, Hong Yeol Yang, Jong Keun Seon

**Affiliations:** 1Department of Biomedical Sciences, Chonnam National University Medical School, Hwasun 58128, Republic of Korea; jaanu.p2206@gmail.com (J.S.); sreesamanvitha95@gmail.com (S.S.K.); 2Department of Orthopaedic Surgery, Center for Joint Disease, Chonnam National University Hwasun Hospital, 322 Seoyang-ro, Hwasun 519763, Republic of Korea; chemokines@naver.com (H.K.K.); jy0194@naver.com (J.Y.K.); 3Korea Biomedical Materials and Devices Innovation Research Center, Chonnam National University Hospital, 42 Jebong-ro, Dong-gu, Gwangju 501757, Republic of Korea

**Keywords:** osteoarthritis, MSC-derived exosomes, anti-inflammation, regenerative medicine, ex vivo OA model

## Abstract

Osteoarthritis (OA) is a prevalent and debilitating joint disorder that affects a substantial proportion of the global population, underscoring the urgent need for therapeutic strategies that extend beyond symptomatic management. Although mesenchymal stem cells (MSCs) have emerged as a promising therapeutic modality, their clinical application remains constrained by several inherent limitations. This study explores a cell-free alternative by investigating the therapeutic potential of exosomes derived from bone marrow (BMSCs), adipose tissue (ADSCs), and umbilical cord (UMSCs) MSCs in mitigating OA pathogenesis, utilizing both in vitro and ex vivo models. Exosomes from each MSC source were isolated and characterized through nanoparticle tracking analysis, transmission electron microscopy, and Western blotting to confirm their identity and purity. Subsequently, their chondroprotective, anti-inflammatory, and regenerative properties were systematically assessed through evaluations of cell viability, expression profiles of inflammatory and chondroprotective markers, and chondrocyte migration assays. The results demonstrate that all three types of MSC-derived exosomes (MSC-Exos) exhibit low cytotoxicity while significantly suppressing proinflammatory markers and enhancing the expression of chondroprotective genes. Notably, BMSC-Exos and UMSC-Exos displayed superior efficacy in attenuating inflammation, promoting cartilage protection, and inhibiting chondrocyte apoptosis. Furthermore, all MSC-Exos markedly enhanced chondrocyte motility, a critical component of cartilage repair. Collectively, these findings support the therapeutic promise of MSC-Exos, particularly those derived from BMSCs and UMSCs, as a targeted, cell-free approach for the treatment of OA compared to ADSCs. By modulating inflammation, promoting cartilage regeneration, and preventing chondrocyte apoptosis, MSC-Exos may serve as a viable and scalable alternative to current MSC-based therapies for this widespread degenerative disease.

## 1. Introduction

Subchondral bone remodeling, synovial inflammation, and the progressive deterioration of articular cartilage are hallmark features of osteoarthritis (OA), a prevalent and disabling degenerative joint disease [1,2]. OA represents a major global cause of pain and disability and imposes substantial economic burdens. As of 2021, approximately 607 million individuals worldwide were affected by OA, marking a considerable increase from the global burden reported in 1990 [3]. In South Korea, the prevalence of OA is similarly alarming. Data from the Korea National Health and Nutrition Examination Survey (KNHANES) have shown that the prevalence of knee OA among Korean adults aged ≥50 years is significant [4]. Specifically, the prevalence of radiographic knee OA was reported to be 35.1%, with a markedly higher prevalence among women (44.3%) compared to men (24.4%) [5]. Current treatment strategies for OA are primarily symptomatic, offering limited efficacy in altering the underlying pathophysiological mechanisms, thereby underscoring the urgent need for novel therapeutic interventions. Mesenchymal stem cells (MSCs) have emerged as promising therapeutic modality for OA due to their robust regenerative and immunomodulatory properties [6]. However, several limitations hinder their widespread clinical application, including high costs, ethical concerns, and the logistical complexity of cell transport and culture expansion [7,8]. To overcome these challenges, recent research has increasingly focused on MSC-derived exosomes (MSC-Exos) as a cell-free therapeutic alternative. MSC-Exos are nanoscale extracellular vesicles that facilitate intercellular communication by delivering a diverse array of bioactive molecules [9]. These vesicles are enriched with proteins, lipids, and nucleic acids that play critical roles in tissue regeneration and cellular signaling [10,11]. Notably, MSC-Exos have been shown to replicate many of the therapeutic effects of their parent MSCs, including the promotion of chondrocyte proliferation, inhibition of apoptosis, attenuation of inflammation, and modulation of immune responses [12]. The therapeutic potential of exosomes derived from various MSC sources, namely adipose-derived stem cells (ADSCs), umbilical cord MSCs (UMSCs), and bone marrow MSCs (BMSCs) has been extensively investigated in the context of OA treatment [13]. ADSC-Exos have demonstrated potent chondrogenic capabilities, enhancing extracellular matrix synthesis and reducing cellular senescence [14]. In addition, they exhibit pronounced anti-inflammatory effects, suppressing chronic inflammation while promoting chondrocyte proliferation and differentiation. UMSC-Exos are particularly noted for their high proliferation capacity and strong immunomodulatory properties. These exosomes have been shown to regulate immune responses, inhibit apoptosis, and facilitate tissue regeneration [15]. Mechanistically, UMSC-Exos promote macrophage polarization toward an anti-inflammatory phenotype, decreasing the proportion of proinflammatory macrophages and enhancing the recruitment of anti-inflammatory subsets. BMSC-Exos, which have been studied extensively, also exert strong chondroprotective and anti-inflammatory effects [16]. These vesicles have been reported to upregulate chondrocyte-specific markers, downregulate catabolic and inflammatory mediators, and inhibit chondrocyte apoptosis. Furthermore, BMSC-Exos can suppress macrophage activation and reduce proinflammatory macrophage infiltration, further supporting their therapeutic efficacy in OA models [17]. Given the heterogeneity in therapeutic outcomes associated with exosomes from different MSC sources, a direct comparative analysis is essential to identify the most efficacious MSC-Exos for OA treatment.

The present study aims to systematically compare the therapeutic potential of exosomes derived from ADSCs, UMSCs, and BMSCs using both in vitro and ex vivo models of OA. By assessing their chondroprotective, anti-inflammatory, and regenerative effects, we aim to elucidate the mechanisms underlying their therapeutic actions and determine the MSC source that confers the most favorable therapeutic profile. This research is expected to provide important insights into the development of MSC-Exo-based cell-free therapies and contribute to the optimization of treatment strategies for this debilitating disease.

## 2. Results

### 2.1. Characterization of BMSC-Exos, ADSC-Exos, and UMSC-Exos Isolated by ATPS

Exosomes were successfully isolated from BMSCs, ADSCs, and UMSCs using the ATPS method. The interfacial layer of ATPS was visualized using Coomassie Brilliant Blue R-250 to stain the PEG phase and layers between the phases (Figure 1A) and constructed a binodal curve for the PEG/Dextran system, calculated phase composition, and characterized the ATPS. The equilibrium phase was measured for volume and density through visually locating the phase separation lines and extracting the bottom phase with a micropipette. (Figure 1E). Characterization of the isolated exosomes was performed using NTA, transmission electron microscopy (TEM), and Western blotting for established exosomal surface markers. NTA demonstrated that exosomes from all three sources exhibited a size distribution characteristic of exosomes. The particle concentrations were determined to be 6.9 × 10^7^ particles/mL for BMSC-Exos, 8.0 × 10^7^ particles/mL for ADSC-Exos, and 1.2 × 10^8^ particles/mL for UMSC-Exos (Figure 1B–D). TEM analysis confirmed the presence of vesicles with the typical cup-shaped morphology in all three exosome groups (Figure 1F–H). The diameters of the exosomes ranged from 30 to 150 nm, aligning with the NTA results. TEM images also demonstrated that the exosomes were well-dispersed and retained their structural integrity following isolation. Western blot analysis revealed the presence of exosomal markers CD63, CD81, and ALIX in all three preparations (Figure 1I), thereby confirming the exosomal identity of BMSC-Exos, ADSC-Exos, and UMSC-Exos.

### 2.2. Cell Cytotoxicity of BMSC-Exos, ADSC-Exos, and UMSC-Exos Using CCK-8 Assay

The cytotoxicity of BMSC-Exos, ADSC-Exos, and UMSC-Exos on chondrocytes was assessed using the CCK-8 assay. Chondrocytes were incubated with increasing concentrations of exosomes for 24 h. The results indicated that chondrocyte viability remained stable at concentrations up to 1000 μg/mL for all three exosome types. No significant reduction in cell viability was observed within this range, indicating that exosomes were well-tolerated by chondrocytes (Figure 2). Notably, a decrease in cell viability was not detected even at higher concentrations, contrary to initial expectations. This consistent trend across all three exosome sources suggests a potential protective or proliferative effect on chondrocytes. Overall, these findings demonstrate that BMSC-Exos, ADSC-Exos, and UMSC-Exos exhibit minimal or no cytotoxicity toward chondrocytes, even at relatively high concentrations. This favorable biocompatibility profile underscores their potential for therapeutic applications in cartilage repair and osteoarthritis treatment.

### 2.3. In Vitro Efficacy of BMSC-Exos, ADSC-Exos, and UMSC-Exos on NF-kB and MAPK Signaling Pathways

To investigate the anti-inflammatory potential of MSC-derived exosomes, we examined their effects on the MAPK and NF-κB signaling pathways in vitro. This analysis focused on the expression of key signaling proteins following stimulation with IL-1β, a well-characterized proinflammatory cytokine. Western blot analysis revealed that treatment with BMSC-Exos, ADSC-Exos, and UMSC-Exos led to a reduction in phosphorylated p65 (pp65) levels compared to IL-1β-stimulated cells. IL-1β exposure alone resulted in elevated pp65 expression, indicative of NF-κB pathway activation; however, the addition of BMSC-Exos, ADSC-Exos, and UMSC-Exos attenuated this response, as reflected by decreased pp65 band intensity. These findings suggest that BMSC-Exos and UMSC-Exos may effectively suppress NF-κB pathway activation in the context of inflammation compared to ADSC-Exos. A similar trend was observed in the MAPK pathway: IL-1β stimulation increased phosphorylation of p38 (pp38), JNK (pJNK), and ERK (pERK), whereas treatment with BMSC-Exos, ADSC-Exos, and UMSC-Exos reduced the expression levels of these phosphorylated proteins compared to IL-1β treatment alone. More enhanced reduction was observed in BMSC-Exos and UMSC-Exos. Collectively, these data indicate that BMSC-Exos and UMSC-Exos may mitigate inflammatory signaling by downregulating both NF-κB and MAPK pathway activation (Figure 3). The observed reductions in pp65, pp38, pJNK, and pERK highlight the therapeutic potential of BMSC-Exos and UMSC-Exos for modulating inflammatory responses. Nevertheless, further studies are warranted to elucidate the specific mechanisms through which BMSC-Exos and UMSC-Exos exert their regulatory effects on MAPK and NF-κB signaling pathways.

### 2.4. In Vitro Efficacy of BMSC-Exos, ADSC-Exos, and UMSC-Exos on Inflammatory Markers

The anti-inflammatory efficacy of BMSC-Exos, ADSC-Exos, and UMSC-Exos were further assessed in vitro via Western blot analysis. Treatment with each type of exosome significantly reduced the expression of proinflammatory protein markers in chondrocytes. Specifically, levels of matrix metalloproteinase-13 (MMP-13), matrix metalloproteinase-3 (MMP-3), interleukin-6 (IL-6), tumor necrosis factor-alpha (TNF-α), and cyclooxygenase-2 (COX-2) were significantly decreased relative to IL-1β treatment. Among the three exosome types, BMSC-Exos and UMSC-Exos demonstrated the most robust effect. All three exosome treatments resulted in the significant downregulation of MMP-13, MMP-3, IL-6, TNF-α, and COX-2 gene expression. These findings collectively underscore the strong anti-inflammatory properties of BMSC-Exos, ADSC-Exos, and UMSC-Exos in vitro, with BMSC-Exos and UMSC-Exos exhibiting the most pronounced efficacy across the assessed inflammatory markers (Figure 4).

### 2.5. In Vitro Efficacy of BMSC-Exos, ADSC-Exos, and UMSC-Exos on Chondroprotective Markers

Concurrently, a marked upregulation of chondroprotective genes, including aggrecan (ACAN) and collagen type II (COL-2), was observed. Among the groups, BMSC-Exos exerted the most pronounced effect, resulting in a significant increase in ACAN and in COL-2 protein levels compared to the control group. Notably, while all exosome types exhibited chondroprotective activity to an extent, BMSC-Exos demonstrated superior efficacy in modulating ACAN and COL-2 expression—key regulators of chondroprotective activity. Collectively, these findings indicate robust chondroprotective effects of BMSC-Exos, ADSC-Exos, and UMSC-Exos in vitro, with BMSC-Exos displaying the highest overall efficacy in enhancing chondroprotective markers (Figure 5).

### 2.6. In Vitro Efficacy of BMSC-Exos, ADSC-Exos, and UMSC-Exos on Apoptotic Marker

The anti-apoptotic potential of BMSC-Exos, ADSC-Exos, and UMSC-Exos was assessed by evaluating caspase-9 protein expression via Western blot analysis (Figure 6). All three exosome types significantly reduced caspase-9 levels compared to untreated controls. Caspase-9 is a key initiator of the intrinsic apoptotic pathway; its activation triggers downstream caspase cascades leading to programmed cell death. Therefore, the observed decrease in caspase-9 levels suggests that exosome treatment effectively inhibits apoptosis in the target cells. This reduction indicates a protective effect of exosomes against IL-1β-induced apoptotic signaling, promoting cell survival and potentially contributing to tissue repair and regeneration.

### 2.7. Chondrocyte Migration Efficacy Using Scratch Wound Assay

Analysis of chondrocyte migration demonstrated significantly enhanced motility following treatment with exosomes derived from different mesenchymal stem cell sources. Scratch wound assays revealed that BMSC-Exos, ADSC-Exos, and UMSC-Exos each significantly promoted chondrocyte migration, with BMSC-Exos showing the most pronounced effect at both 24 h and 36 h. Chondrocytes cultured with exosomes at a concentration of 5 µg/mL exhibited greater proliferation compared to the control group. The enhanced migration observed in chondrocytes following exosome treatment is likely due to the bioactive molecules carried by exosomes, such as growth factors, cytokines, and microRNAs. These exosomal components can modulate cellular signaling pathways involved in cytoskeletal remodeling, cell adhesion, and motility (Figure 7).

### 2.8. H&E Staining Reveals Cartilage Histopathology

Histological analysis using H&E staining demonstrated substantial structural alterations in cartilage following IL-1β stimulation. Cartilage from the untreated control group exhibited normal architecture, with an intact matrix and orderly chondrocyte distribution. In contrast, samples exposed to IL-1β showed pronounced degenerative features, including diminished cartilage thickness, disrupted chondrocyte organization, increased cellularity in the superficial layer, and surface irregularities. Administration of BMSC-Exos, ADSC-Exos, and UMSC-Exos conferred varying degrees of histological protection against IL-1β-induced damage. All exosome treatments exerted some inhibitory effects on cartilage degeneration, with more robust regenerative responses observed in the ADSC and UMSC exosome groups, particularly at days 7 and 14 (Figure 8). Nonetheless, some treated specimens continued to display signs of degeneration, such as chondrocyte loss, reduced matrix staining, and superficial fibrillation. These observations indicate that although exosome-based interventions may attenuate IL-1β-mediated cartilage injury, the reparative process remains complex and may necessitate further optimization of therapeutic protocols to achieve complete tissue restoration.

### 2.9. Safranin O/Fast Green Staining Reveals Proteoglycan Distribution in Cartilage Samples

Safranin O/fast green staining of paraffin-embedded cartilage sections revealed clear differences in proteoglycan content among the experimental groups over time (Figure 9). In the control group, intense and uniform Safranin O staining was observed at all time points (3, 7, and 14 days), indicating well-preserved proteoglycan levels in the cartilage matrix. In contrast, cartilage exposed to IL-1β alone showed a marked reduction in Safranin O staining, particularly evident by Day 7 and further pronounced by Day 14, reflecting significant proteoglycan depletion characteristic of osteoarthritic degeneration. Notably, treatment with exosomes derived from BMSCs, ADSCs, and UMSCs mitigated the loss of proteoglycans induced by IL-1β. All exosome-treated groups exhibited stronger Safranin O staining compared to the IL-1β group at both 7 and 14 days, indicating better preservation of cartilage matrix. However, a gradual decrease in staining intensity was still observed in all groups over time, likely due to the release of glycosaminoglycans (GAGs) into the culture supernatant during prolonged ex vivo incubation. These findings confirm that IL-1β accelerates proteoglycan loss in cartilage, while exosome treatments, particularly from BMSCs, help maintain matrix integrity and can delay proteoglycan (GAG) loss in the ex vivo OA model; however, they do not completely prevent it, especially by day 14. Safranin O staining intensity reliably reflected changes in proteoglycan content, supporting its value for assessing cartilage degeneration and the efficacy of chondroprotective interventions.

### 2.10. Picrosirius Red Staining Reveals Collagen Content in Cartilage Samples

Collagen, a primary component of the extracellular matrix, imparts tensile strength and structural integrity to cartilage tissue. Type II collagen is especially prevalent in knee cartilage and connective tissues. Picrosirius red staining, which leverages collagen’s birefringence under polarized light, enables visualization of collagen networks. In ex vivo models, IL-1β exposure commonly reduces collagen content. However, treatment with BMSC-Exos, ADSC-Exos, and UMSC-Exos over days 3 and 7 appeared to prevent this decline, attenuating cartilage degradation and preserving collagen levels. By day 14, collagen content was notably restored in exosome-treated groups, suggesting inhibition of IL-1β activity and support of matrix repair. These results imply that MSC-derived exosomes may exert a protective effect on collagen integrity within cartilage tissue (Figure 10).

## 3. Discussion

Stem cell-derived exosomes represent a paradigm shift in regenerative therapeutic strategies. Numerous studies have assessed their therapeutic potential in both in vitro and in vivo models, yielding promising outcomes [18]. To enable successful clinical translation, the therapeutic application of exosomes must satisfy stringent quality and safety criteria. While much of the research in regenerative medicine has focused on MSC-Exos, their production typically involves in vitro or ex vivo expansion processes, which require careful optimization to ensure safety and compliance with industrial and regulatory standards. Emerging evidence suggests that BMSC-Exos and UMSC-Exos may offer enhanced therapeutic efficacy compared to ADSC-Exos. This is supported by superior performance in in vitro and ex vivo inflammatory osteoarthritis (OA) models utilizing human cartilage explants, wherein BMSC-Exos and UMSC-Exos more effectively attenuated disease progression. A recent investigation reported that exosomes derived from human embryonic MSCs significantly promoted cartilage repair and chondrocyte proliferation in a rat OA model. Weekly intra-articular administration of these exosomes resulted in complete restoration of hyaline cartilage and subchondral bone within 12 weeks, closely approximating native tissue architecture. In contrast, control groups treated with PBS exhibited only fibrous repair tissue [19]. In OA, excessive inflammation is a key contributor to cartilage degradation and impaired tissue regeneration. Proinflammatory cytokines such as TNF-α, IL-1β, and IL-6 are consistently elevated in OA and are known to drive cartilage breakdown through NF-κB/MAPK signaling and upregulation of matrix-degrading enzymes. Targeting this inflammatory milieu, either by enhancing anti-inflammatory cytokines (e.g., IL-4, IL-10, TGF-β) or directly inhibiting proinflammatory mediators (e.g., TNF-α), offer promising therapeutic avenues for OA management [20]. Notably, a study demonstrated that exosomes derived from inflamed synovial fibroblasts induced osteoarthritic changes in chondrocytes. These exosomes mediated intercellular signaling between synovial fibroblasts and articular chondrocytes, contributing to OA-like changes and further aggravating cartilage degradation. This underscores the complex interplay between synovial inflammation and chondrocyte dysfunction in OA pathogenesis [21]. MicroRNAs (miRNAs) encapsulated within exosomes are emerging as critical regulators of OA pathophysiology through their roles in intercellular communication. These non-coding RNAs influence gene expression related to inflammation, cartilage catabolism, and tissue regeneration [22]. Exosomes function as stable carriers for miRNAs, enabling targeted delivery to chondrocytes and synovial fibroblasts, which are central to OA progression. A pivotal study demonstrated that exosomes derived from miR-140-5p-overexpressing synovial mesenchymal stem cells (SMSC-140-Exos) markedly enhanced cartilage regeneration and mitigated OA progression in rat models. These exosomes promoted chondrocyte proliferation and migration without compromising ECM production, a key determinant of cartilage integrity [23].

Exosomal non-coding RNAs (ncRNAs) have emerged as critical regulators in the pathogenesis and treatment of bone-related disorders, including osteoporosis (OP), osteoarthritis (OA), and impaired fracture healing. These molecules mediate intercellular communication, influence bone cell activity, and regulate inflammatory and metabolic pathways, thereby presenting novel therapeutic possibilities [24]. In ex vivo models, inflammatory conditions are simulated via cytokine stimulation (e.g., IL-1β, TNF-α) in cartilage explants, recapitulating hallmark features of OA such as extracellular matrix degradation and chondrocyte dysfunction [25]. The current study employed these models to assess the therapeutic efficacy of MSC-Exos, comparing outcomes across exosomes derived from BMSCs, ADSCs, and UMSCs. Findings indicated that MSC-Exos, at a single dose of approximately 5 μg/mL in 2D cultures and 50 μg/mL in the ex vivo OA model, significantly attenuated cytokine-induced inflammation and cartilage degradation, with efficacy varying by cell source. Notably, BMSC-Exos and UMSC-Exos exhibited greater suppression of proinflammatory cytokine expression compared to ADSC-Exos, potentially due to differences in cargo composition, including specific miRNAs and growth factors. Clinically, intra-articular injection of exosomes has emerged as an alternative to PDRN or corticosteroid treatments for OA. Nevertheless, consistent with prior studies [26,27,28], MSC-Exos led only to partial rescue of inflamed chondrocytes, reducing proinflammatory mediators and catabolic enzymes while concurrently demonstrating time-dependent adverse effects on matrix synthesis and collagen production. Moreover, the beneficial effects of MSC-Exos on inflamed chondrocytes were found to be transient. MSC-derived exosomes mitigate cellular senescence through multiple interconnected mechanisms that collectively enhance cellular rejuvenation and promote tissue regeneration. These vesicles carry a variety of bioactive molecules capable of modulating gene expression in recipient cells and activating pathways involved in cell cycle progression and tissue repair [29,30].

MSC exosomes mitigate oxidative stress by delivering antioxidant enzymes and enhancing the expression of antioxidant genes, thereby reducing ROS levels and alleviating DNA damage [31]. This attenuation of oxidative stress contributes to the preservation of mitochondrial function and cellular homeostasis. In addition, MSC exosomes modulate signaling pathways associated with senescence, including the p53 pathway. Through the transfer of miRNAs and proteins capable of suppressing p53 activation, these exosomes facilitate cell cycle progression and inhibit senescence-associated growth arrest. MSC exosomes also attenuate the senescence-associated secretory phenotype (SASP) by downregulating proinflammatory cytokines such as IL-6 and CCL7, thereby fostering a microenvironment conducive to cellular rejuvenation. Moreover, MSC exosomes enhance DNA repair pathways and reduce DNA damage accumulation, as evidenced by decreased γ-H2AX–positive cells following exosome treatment [32,33], thus supporting genomic stability and delaying cellular senescence. By delivering a range of bioactive molecules, MSC exosomes have also been shown to promote chondrocyte proliferation and inhibit apoptosis via the lncRNA-KLF3-AS1/miR-206/GIT1 axis. This pathway involves the transfer of lncRNA KLF3-AS1 to chondrocytes, which modulates miR-206 activity and regulates GIT1 expression. Upregulation of GIT1 subsequently enhances chondrocyte survival and proliferation while suppressing apoptotic processes [34]. Furthermore, ex vivo comparisons between MSC-derived exosomes and their parental cells revealed that exosomes replicate many of the therapeutic properties of intact MSCs, including the downregulation of ADAMTS-5, inhibition of NF-κB signaling, and maintenance of collagen II and aggrecan synthesis. A study employing an ex vivo equine cartilage explant model utilized metabolomic analysis to investigate OA pathogenesis [35]. Although the specific application of exosomes was not described in that study, the model is well-suited for assessing the impact of exosome-mediated interventions on cartilage degradation and repair under controlled conditions, free from systemic confounders. In a collagenase-induced OA murine model, exosomes derived from BMMSCs attenuated cartilage degradation by upregulating type II collagen and aggrecan while suppressing MMP-13 and ADAMTS-5 expression in chondrocytes [9]. Emerging evidence supports the therapeutic potential of MSC-derived exosomes in alleviating OA through anti-inflammatory activity, cartilage regeneration, and matrix homeostasis restoration. Preconditioning MSCs with TNF-α may enhance exosome yield and enrich beneficial miRNAs, such as miR-100-5p, which modulates chondrocyte function by inhibiting mTOR signaling [36]. Similarly, umbilical cord MSC exosomes exerted anti-inflammatory effects and suppressed subchondral bone remodeling in OA models [37,38]. In our current study, we compared the efficacy of BMSC-, ADSC-, and UMSC-derived exosomes in suppressing IL-1β-induced activation. All exosome types demonstrated inhibitory effects, although with varying magnitudes (Figure 11). Future research should aim to refine the production and delivery of MSC-derived exosomes, particularly BMSC- and UMSC-derived exosomes, to optimize their therapeutic efficacy. Additionally, elucidating the molecular mechanisms underlying the enhanced performance of BMSC- and UMSC-derived exosomes may facilitate the development of improved exosome-based therapies from diverse cellular sources.

### Challenges and Future Directions

Although recent studies have reported promising results for MSC-derived exosomes in the treatment of osteoarthritis (OA) using in vitro and ex vivo models, several challenges must be addressed before their widespread clinical application. Notably, the lack of in vivo validation remains a critical gap that future investigations will address by exploring the exosomal cargo composition in greater detail. Optimizing the production, isolation, and delivery methods particularly through the ATPS isolation technique is essential to fully harness their therapeutic potential. Furthermore, elucidating the specific molecular mechanisms underlying the enhanced efficacy of exosomes derived from bone marrow and umbilical cord MSCs is necessary, as this knowledge could inform strategies to improve the therapeutic performance of exosomes from alternative sources [39,40,41].

## 4. Materials and Methods

### 4.1. Antibodies and Reagents

Dulbecco’s Modified Eagle’s Medium (DMEM; 1×), penicillin–streptomycin (pen-strep), and fetal bovine serum (FBS) were obtained from Gibco (Thermo Fisher Scientific, Waltham, MA, USA). Human recombinant IL-1β was purchased from R&D Systems (Minneapolis, MN, USA) and reconstituted in phosphate-buffered saline (PBS) containing 0.5% bovine serum albumin (BSA). Antibodies against type II collagen (COL-2), COX-2, MMP-13, p65, phosphorylated p65 (pp65), ERK, phosphorylated ERK (pERK), p38, phosphorylated p38 (pp38), JNK, phosphorylated JNK (pJNK), CD63, CD81, and ALIX were procured from the following suppliers: Abcam (Boston, MA, USA) for COL-2 and COX-2; Bioss (Woburn, MA, USA) for MMP-13; Cell Signaling Technology (Danvers, MA, USA) for p65, pp65, ERK, pERK, p38, pp38, JNK, pJNK, CD63, and CD81; and Santa Cruz Biotechnology (Dallas, TX, USA) for ALIX. Goat anti-mouse IgG secondary antibody was purchased from ZyMax (Thermo Fisher Scientific), and a conjugated goat anti-rabbit IgG (H + L) secondary antibody was obtained from Novex Life Technologies (Thermo Fisher Scientific, Waltham, MA, USA).

### 4.2. Source of Chondrocytes and MSCs Culture

Human articular chondrocytes, BMSCs, and ADSCs were obtained from ScienCell (catalog numbers 4650, 7500, and 7510, respectively, Carlsbad, CA, USA). Umbilical cord-derived mesenchymal stem cells (UMSCs) were provided by MEDIPOST Co., Ltd. Chondrocytes, BMSCs, and ADSCs were cultured in DMEM, whereas UMSCs were maintained in alpha-Minimum Essential Medium (α-MEM). All media were supplemented with 10% heat-inactivated FBS and 1% pen-strep. Cells were incubated at 37 °C in a humidified atmosphere containing 5% CO_2_. Once cultures reached approximately 80% confluence, cells were passaged; cells between passages 3 and 5 were used for all experiments. For exosome isolation, mesenchymal stem cells at passages 3 or 4 were cultured in DMEM supplemented with 10% heat-inactivated exosome-free serum and 1% pen-strep under identical incubation conditions. Donor criteria for human articular chondrocytes were not disclosed by the supplier; researchers are encouraged to contact ScienCell directly for detailed characterization information.

### 4.3. Exosome Isolation from BMSC, ADSC, and UMSCs

Mesenchymal stem cells (BMSCs, ADSCs, and UMSCs) were cultured in 100 mm dishes at a seeding density of 1 × 10^6^ cells per well. The cells were maintained in DMEM supplemented with 10% exosome-depleted fetal bovine serum and 1% penicillin–streptomycin at 37 °C in a humidified atmosphere containing 5% CO₂. Culture supernatants were collected every 48 h and stored for subsequent exosome isolation. Exosomes were isolated using differential centrifugation, beginning with centrifugation at 5000× *g* for 5 min, followed by 7000–8000× *g* for 20 min at 4 °C. The resulting pellet was further purified using an aqueous two-phase system (ATPS) composed of Polyethylene Glycol (PEG) and Dextran (DEX) at a 1:1 volume ratio. The mixture was incubated overnight at 4 °C. Phase separation was achieved by centrifugation at 5000× *g* for 5 min. The exosome-rich DEX phase was repeatedly washed with ATPS solution until a clear, non-viscous interface was observed. Purified exosomes were then resuspended in phosphate-buffered saline, flash-frozen, and stored at −80 °C prior to lyophilization. Lyophilized exosome preparations were stored at −80 °C until further use.

### 4.4. Characterization of the ATPS and Exosomes

Characterization of the PEG/DEX aqueous two-phase system was performed by constructing a binodal curve to determine phase compositions. The equilibrium phases were evaluated by measuring volume and density, with the bottom phase extracted via micropipette following visual identification of the phase boundary. To facilitate visualization of the ATPS structure, Coomassie Brilliant Blue R-250 was used to stain the PEG phase and interfacial layers. Exosome size distribution and concentration were analyzed using the ZetaView^®^ nanoparticle tracking analysis (NTA) system (Particle Metrix GmbH, Meerbusch, Germany) in accordance with the manufacturer’s instructions. For morphological analysis, exosomes were adsorbed onto 2 nm copper grids and examined using a JEOL JEM-2100F field emission transmission electron microscope (FE-TEM) (JEOL Ltd., Akishima, Tokyo, Japan), enabling visualization of BMSC-Exos, ADSC-Exos, and UMSC-Exos. Western blotting was conducted to confirm exosome identity by detecting characteristic exosomal markers, including CD63, CD81, and ALIX.

### 4.5. Role of Exosomes in IL-1β-Induced Chondrocyte Inflammation In Vitro

Human articular chondrocytes were cultured in 65 mm dishes at a density of 5 × 10⁵ cells/well in DMEM supplemented with 10% FBS and 1% penicillin–streptomycin under standard conditions (37 °C, 5% CO_2_). After 24 h of incubation and confirmation of confluency, inflammation was induced by replacing the culture medium with serum-free exosome medium containing 10 ng/mL interleukin-1β (IL-1β), a concentration previously validated to simulate osteoarthritis-like cellular responses in our previous study [42]. Lyophilized exosomes (BMSC-Exos, ADSC-Exos, UMSC-Exos) were reconstituted in phosphate-buffered saline to yield 10 mg/mL stock solutions, and working concentrations were standardized to 5 µg/mL based on prior reports [42]. The timing of post-inflammation exosome treatment (24 h) was selected to emulate clinical intervention paradigms rather than preventive administration. Following a 24 h treatment period with exosomes, cellular proteins were extracted using radioimmunoprecipitation assay (RIPA) buffer containing 0.5% sodium deoxycholate and 1% NP-40, freshly supplemented with protease and phosphatase inhibitor cocktails. This extraction approach facilitates efficient recovery of total cellular proteins while preserving biological activity for subsequent analyses.

### 4.6. Cellular Cytotoxicity Using the CCK-8 Assay

Cell viability was assessed using the CCK-8 assay kit (Dojindo Molecular Technologies, Kumamoto, Japan) following BMSC-Exos, ADSC-Exos, and UMSC-Exos treatment. Cells were seeded in 96-well plates at a density of 1 × 10^4^ cells per well and allowed to adhere for 24 h. Subsequently, the cells were exposed to BMSC-Exos, ADSC-Exos, and UMSC-Exos at concentrations ranging from 1000 µg/mL to 1 µg/mL for an additional 24 h. After treatment, 10 µL of CCK-8 reagent was added to each well and incubated for 2 h at room temperature. Absorbance at 450 nm was measured using a BioTek Synergy HTX multimode plate reader (Agilent Technologies, Winooski, VT, USA) to quantify the formazan dye produced, which correlates with the number of viable cells. The CCK-8 assay is a convenient, ready-to-use colorimetric method that evaluates cell viability and cytotoxicity by detecting the metabolic activity of living cells. Viable cells reduce the WST-8 reagent to a water-soluble, orange formazan product, allowing for rapid and sensitive quantification. Compared to other tetrazolium-based assays such as MTT, XTT, and MTS, CCK-8 offers higher sensitivity and does not require additional solubilization steps, making it especially suitable for high-throughput and routine cell viability measurements

### 4.7. Whole-Cell Lysate Preparation and Wesatern Blot Analysis

Following treatment, cells were lysed in RIPA buffer supplemented with protease and phosphatase inhibitors. Total protein concentrations in the resulting whole-cell lysates were determined using a bicinchoninic acid (BCA) assay kit (Thermo Scientific, Waltham, MA, USA). Protein standards and samples were dispensed into a 96-well plate, mixed with a working reagent, and incubated at 37 °C for 30 min prior to absorbance measurement at 562 nm. Protein aliquots (8 µg per sample) were heat-denatured at 95 °C for 5 min, resolved by SDS-PAGE, and transferred onto PVDF membranes. Membranes were blocked with 5% non-fat milk (DifcoTM, Becton Drive, Franklin Lakes, NJ, USA) for 90 min at 4 °C, followed by overnight incubation with primary antibodies targeting COX-2, COL-2, MMP-13, p65, pp65, ERK, pERK, p38, pp38, JNK, pJNK, c-Jun, pc-Jun, and GAPDH. After washing with TBST, membranes were incubated with horseradish peroxidase-conjugated secondary antibodies (1:5000 dilution) for 2 h at 4 °C. Subsequent TBST washes were followed by treatment with chemiluminescent substrates. Protein bands were visualized using enhanced chemiluminescence (ECL) and quantified using ImageJ software (Version 1.54j, National Institutes of Health, Bethesda, MD, USA).

### 4.8. Scratch Wound Assay

To assess cell migration, a scratch wound healing assay was performed. Chondrocytes were cultured to confluence in 6-well plates. A linear scratch approximately 1 mm in width was created across the monolayer using a 200 μL pipette tip. Wells were gently rinsed twice with PBS to remove non-adherent cells. The control group received fresh medium alone, whereas the experimental groups were treated with 5 µg/mL of BMSC-Exos, ADSC-Exos, or UMSC-Exos. Plates were incubated at 37 °C in a humidified atmosphere containing 5% CO_2_. Micrographs of the scratch region were captured at 0, 24, and 36 h post-injury using an inverted microscope equipped with a digital camera. The extent of cell migration into the denuded area was quantified through image analysis at each time point.

### 4.9. Experimental Design for the Establishment of an Ex Vivo OA Model

Human cartilage specimens were obtained during knee surgery for use in an ex vivo osteoarthritis model. Cartilage explants were excised, rinsed in PBS containing 1% penicillin–streptomycin, and manually trimmed to a diameter of 6–8 mm using a surgical blade (including both articular cartilage and subchondral bone) were excised (Figure 12A). Samples were cultured in chondrocyte-specific medium (DMEM/F12 supplemented with 1% penicillin–streptomycin) for 3 days at 37 °C in a 5% CO_2_ incubator to preserve tissue viability and integrity (Figure 12B,C). On day 0, after confirming sterility, osteoarthritic conditions were induced by supplementing the medium with 10 ng/µL IL-1β, excluding the control group. Explants were allocated into five groups: (1) control, (2) OA-mimicking (IL-1β only), (3) BMSC-Exos treatment, (4) ADSC-Exos treatment, and (5) UMSC-Exos treatment. A concentration of 50 μg/mL exosomes was applied in the treatment groups to account for the reduced permeability inherent to three-dimensional tissue matrices. Media containing inflammatory stimuli and treatments were refreshed every 3 days for either 7 or 14 days. At the end of the treatment period, cartilage explants were fixed, sectioned, and subjected to histological examination to assess structural and morphological alterations.

### 4.10. Sampling and Tissue Preparation

Human cartilage specimens were fixed in 4% paraformaldehyde for 24 h, followed by decalcification in EDTA for 2–3 days. The samples were subsequently dehydrated through a graded ethanol series and embedded in paraffin according to standard procedures. Sagittal sections with a thickness of 5 μm were obtained from knee tissue. To assess tissue structure and pathological features, sections were stained with hematoxylin and eosin (H&E), Safranin O/fast green, and picrosirius red. Safranin O/fast green staining was used to assess proteoglycan content in articular cartilage, as the intensity of Safranin O staining is indicative of proteoglycan levels. Picrosirius red staining was performed according to the manufacturer’s instructions (Abcam 150681) to visualize collagen content. This method specifically binds to collagen fibrils, enabling the differentiation of various collagen types.

### 4.11. Data Analysis

All statistical analyses were performed using GraphPad Prism version 9.0 (GraphPad Software, San Diego, CA, USA). For parametric data involving comparisons among multiple groups, a one-way or two-way analysis of variance (ANOVA) was conducted, followed by Tukey’s post hoc test for multiple comparisons. Results are presented as mean ± standard deviation (SD), with all experiments conducted in triplicate (n = 3). A *p*-value of < 0.05 was considered statistically significant. Statistical significance is indicated as follows: ns (not significant), * (*p* < 0.05), ** (*p* < 0.01), *** (*p* < 0.001), and **** (*p* < 0.0001), relative to the control group unless otherwise specified.

## 5. Conclusions

In conclusion, our study presents compelling evidence supporting the therapeutic potential of MSC-Exos in mitigating OA pathogenesis. By demonstrating their capacity to downregulate proinflammatory markers, upregulate chondroprotective gene expression, and enhance chondrocyte motility in both in vitro and ex vivo models, we highlight the promise of these nanoscale vesicles as a targeted therapeutic approach for OA. Notably, BMSC-Exos and UMSC-Exos consistently showed superior efficacy in reducing inflammation, preserving cartilage integrity, and inhibiting apoptosis, highlighting their potential as a particularly effective cell-free intervention compared to ADSC-Exos in vitro. In contrast, BMSC-Exos exhibited the most pronounced therapeutic benefits, surpassing both ADSC-Exos and UMSC-Exos in their ability to protect tissue in ex vivo models. These findings support the need for further research focused on optimizing the application, delivery, and clinical translation of MSC-Exos especially those derived from BMSCs as a novel therapeutic strategy for this prevalent and debilitating condition.

## Figures and Tables

**Figure 1 ijms-26-05447-f001:**
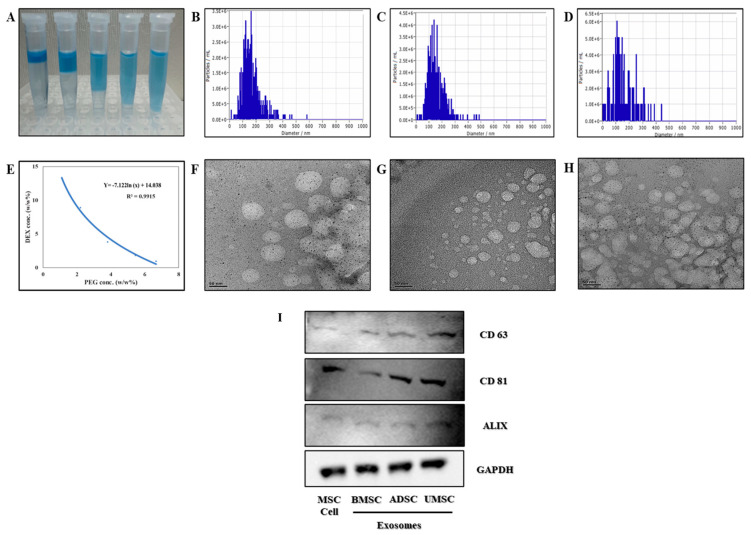
(**A**) Aqueous two-phase system (visualized by adding Coomassie Brilliant Blue R-250). (**B**–**D**) The size distribution of BMSC-Exos, ADSC-Exos, and UMSC-Exos, as determined by nanoparticle tracking analysis. (**E**) Phase diagram of PEG/DEX ATPS. The two-phase forms when system concentration is above the binodal curve. (**F**–**H**) Transmission electron microscopy images of BMSC-Exos, ADSC-Exos, and UMSC-Exos. (**I**) Western blot expression of the exosomal markers CD81, CD63, and ALIX.

**Figure 2 ijms-26-05447-f002:**
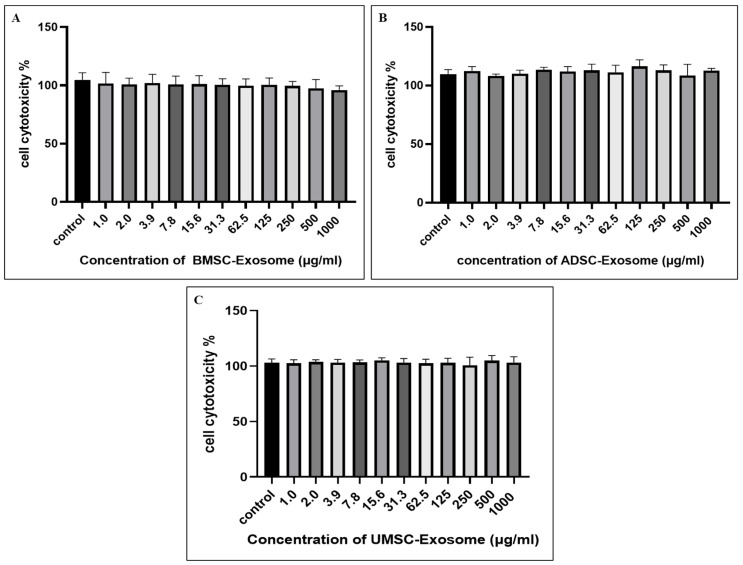
(**A**) Cell viability as a function of BMSC-Exos. (**B**) Cell viability as a function of ADSC-Exos. (**C**) Cell viability as a function of UMSC-Exos—all in decreasing concentrations from 1000 μg/mL to 1.0 μg/mL. Data are shown as the mean ± standard deviation (n = 3). The data depicted represent significance at *p* < 0.05 in all data sets.

**Figure 3 ijms-26-05447-f003:**
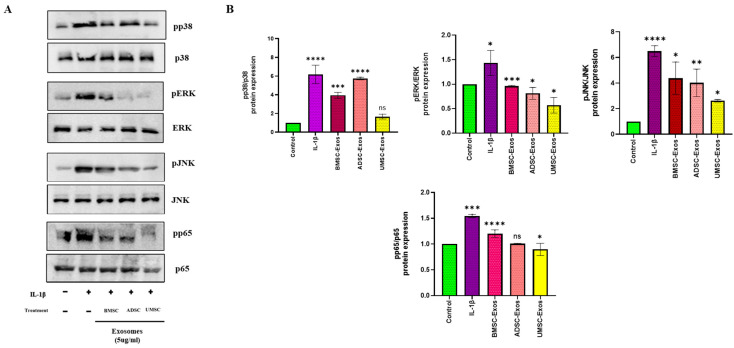
Effects of BMSC-Exos, ADSC-Exos, and UMSC-Exos on MAPK pathways (p38, pERK, and pJNK) and NF-κB (pp65) in chondrocytes. (**A**) Protein expression levels of BMSC-Exos, ADSC-Exos, and UMSC-Exos on IL-1β treated chondrocytes and (**B**) quantitative analysis. Data are presented as the mean ± standard deviation (n = 3). ns = non-significant, * *p* < 0.05, ** *p* < 0.01, *** *p* < 0.001, and **** *p* < 0.0001 compared with the control group.

**Figure 4 ijms-26-05447-f004:**
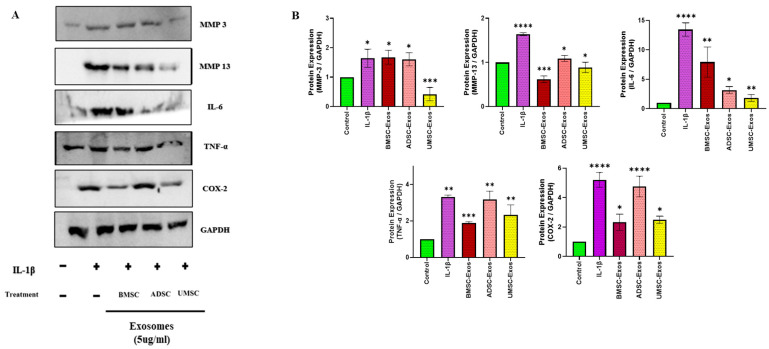
Effects of BMSC-Exos, ADSC-Exos, and UMSC-Exos on inflammation mediators (**A**) Protein expression levels of MMP-3, MMP-13, IL-6, TNF-α, and COX-2 detected via Western blot analysis and (**B**) quantitative analysis. Data are presented as the mean ± standard deviation (n = 3). * *p* < 0.01, ** *p* < 0.05, *** *p* < 0.001 and **** *p* < 0.0001 compared to the control group.

**Figure 5 ijms-26-05447-f005:**
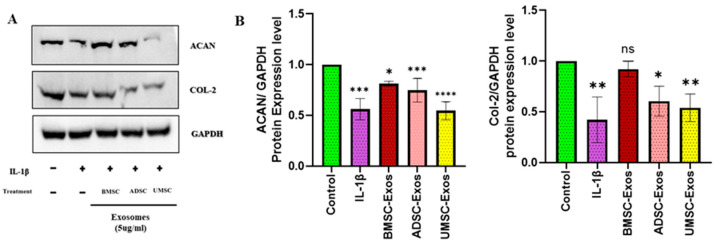
Effects of BMSC-Exos, ADSC-Exos, and UMSC-Exos on chondroprotective markers. (**A**) Protein expression levels and quantitative analysis of ACAN and COL-2 were found through Western blot analysis and (**B**) quantitative analysis. Data are presented as the mean ± standard deviation (n = 3). ns = Non significant, * *p* < 0.01, ** *p* < 0.05, *** *p* < 0.001 and **** *p* < 0.0001 compared to the control group.

**Figure 6 ijms-26-05447-f006:**
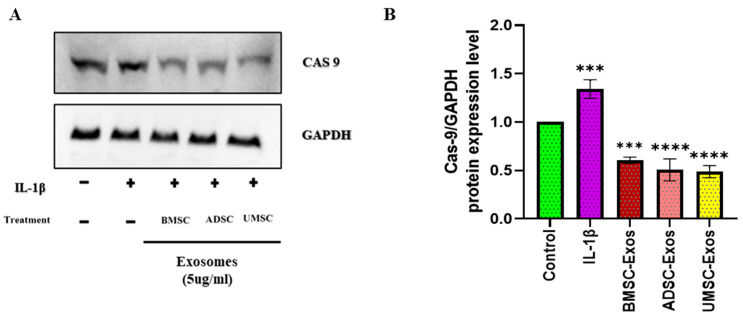
Effects of BMSC-Exos, ADSC-Exos, and UMSC-Exos on apoptotic markers (**A**) Protein expression levels and quantitative analysis of Cas-9 were found through Western blot analysis and (**B**) quantitative analysis. Data are presented as the mean ± standard deviation (n = 3). *** *p* < 0.001 and **** *p* < 0.0001 compared to the control group.

**Figure 7 ijms-26-05447-f007:**
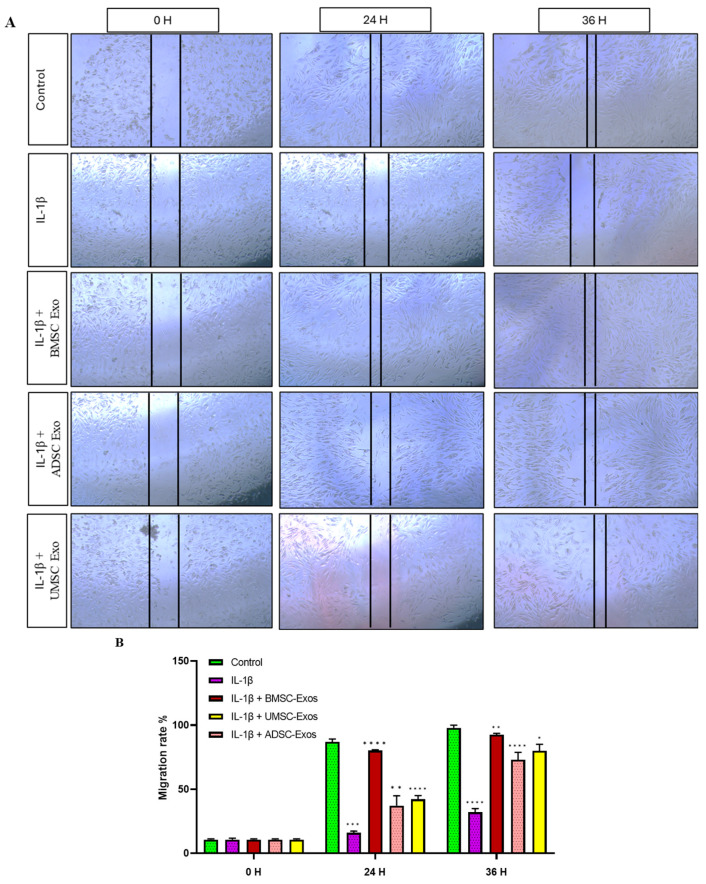
(**A**) Migration rate in an in vitro model of osteoarthritis. The scratch wound assay demonstrates the migration rates of osteoarthritic chondrocytes treated with IL-1β (10 ng/mL) and various exosomes (BMSC-Exos, ADSC-Exos, and UMSC-Exos). Scale bar = 100 μm. (**B**) followed by migration rate % quantitative analysis. BMSC-Exos (5 μg/mL) significantly promoted the migration of osteoarthritis chondrocytes induced by IL-1β compared with control group at both 24 h and 36 h. Data are presented as the mean ± standard deviation (n = 3). * *p* < 0.01, ** *p* < 0.05, *** *p* < 0.001 and **** *p* < 0.0001 compared to the control group.

**Figure 8 ijms-26-05447-f008:**
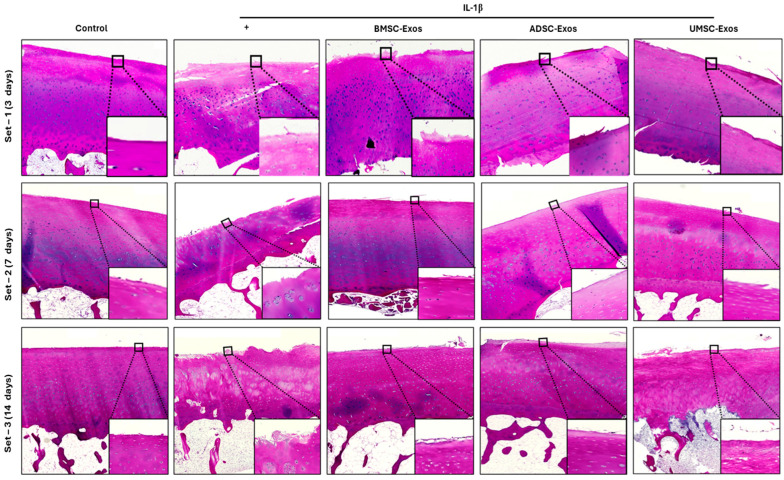
H&E staining of ex vivo model osteoarthritis. Tissues were pretreated with IL-1β (10 ng/mL) and various exosomes (BMSC-Exos, ADSC-Exos, and UMSC-Exos), depicting irregularity in the cartilage structure.

**Figure 9 ijms-26-05447-f009:**
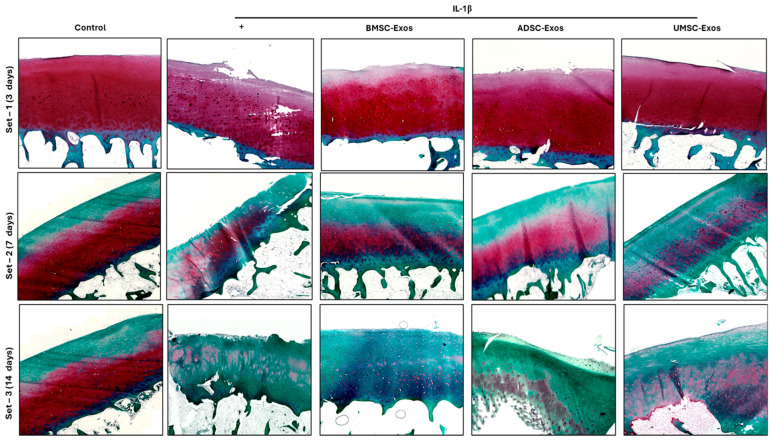
Safranin O staining of ex vivo model of osteoarthritis. Tissues were pretreated with IL-1β (10 ng/mL) and various exosomes (BMSC-Exos, ADSC-Exos, and UMSC-Exos), depicting the proteoglycan GAG content in the tissues. Red color represents the GAG distribution and green stains the cytoplasm.

**Figure 10 ijms-26-05447-f010:**
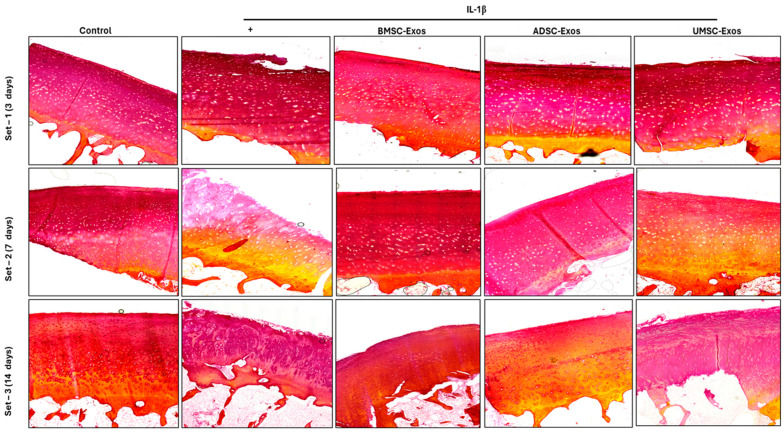
Collagen level of cartilage tissue evaluated by Picrosirius Red assay. Tissues were pretreated with IL-1β (10 ng/mL) and various exosomes (BMSC-Exos, ADSC-Exos, and UMSC-Exos). Collagen distribution is depicted in red, with pink areas indicating reduced collagen content. Yellow regions highlight cytoplasm and muscle fibers. Scale bar = 100 μm.

**Figure 11 ijms-26-05447-f011:**
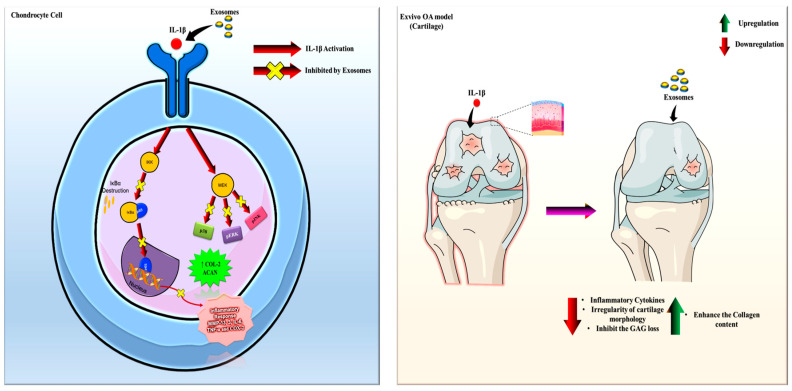
Schematic representation of the in vitro and ex vivo models of osteoarthritis.

**Figure 12 ijms-26-05447-f012:**
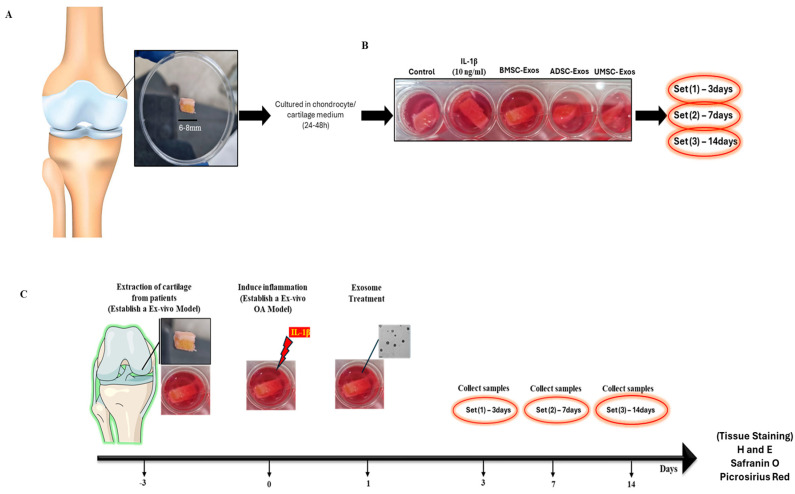
Ex vivo inflammatory OA model using human knee osteochondral specimens. (**A**) Initial cartilage diameter measurement (6–8 mm) with a cartilage thickness of (1–3 mm) of day 0 specimens. (**B**) Experimental groups: control, IL-1β-stimulated, and various exosome treatments. (**C**) Timeline of the ex vivo inflammatory OA model experiment.

## Data Availability

All data generated and analyzed during the current study are available.

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
