# Peer review of "Comparative Efficacy of Exosomes Derived from Different Mesenchymal Stem Cell Sources in Osteoarthritis Models: An In Vitro and Ex Vivo Analysis"

_ijms, 2025, doi:10.3390/ijms26125447_

Round 1

Reviewer 1 Report

Comments and Suggestions for Authors

In this article, the authors explored the use of mesenchymal stem cell exosomes as potential therapeutic candidates for osteoarthritis. They extracted exosomes from bone MSCs, adipose MSCs, and umbilical cord MSCs. After characterizing the exosomes, they applied them as treatments in vitro and ex vivo OA injury models and measured their impact on inflammatory and chondroprotective markers. The results show the potential for using exosomes to treat or slow down the progression of OA.

I have the following questions and comments for the authors:

(1.) In section 2.1 (lines 98 to 99, page 3), the authors mention “Exosomes were successfully isolated from BMSCs, ADSCs, and UMSCs using the 98 ATPS method (Figure 2A, E).”. And the caption for Figure 2E (page 3) says “The aqueous two-phase forms when the system concentration is above the binodal curve. Phase diagram of PEG/DEX ATPS.”. However, it is not clear what figure 2E actually shows. It does not look like a binodal curve since it doesn’t show the composition of both phases. It is also not clear if it shows the composition of one of the phases in your experiments. Please elaborate on the details in the results section, and also please include a clearer caption.

(2.) All the figures have very small font size for the text, and the figures are very low resolution (making the text hard to read even when zooming in). Please increase both the font size and the resolution of the figures.

(3.) In section 2.2 (lines 119 to 120, page 3), the authors mention “The cytotoxicity of BMSC-Exos, ADSC-Exos, and UMSC-Exos on chondrocytes was assessed using the CCK-8 assay.” However, there is no description of this assay in the Methods section. Please include a section in the Methods to describe how this assay was performed.

(4.) The text in section 2.3 (lines 137 to 155, page 4) discusses the results for BMSC and UMSC exos, but not the results for ADSC exos (which are shown in the accompanying figure 4). Please update the text to include the results for ADSC exos.

(5.) In sections 2.4, 2.5, and 2.6 (pages 5 and 6), why were western blots used for quantifying proteins instead of ELISA assays? ELISA-based quantification is more reliable than trying to quantify based on western blot images, and customizable multiplex ELISA assays are available which allow quantifying multiple proteins simultaneously.

(6.) In section 2.4 (page 5, lines 163 to 167), the authors mention “Treatment with each type of exosome significantly reduced the expression of proinflammatory protein markers in chondrocytes. Specifically, levels of matrix metalloproteinase-13 (MMP-13), matrix metalloproteinase-3 (MMP-3), interleukin-6 (IL-6), tumor necrosis factor-alpha (TNF-α), and cyclooxygenase-2 (COX-2) were significantly decreased relative to untreated controls (p < 0.05).” However, looking at the accompanying Figure 5, most of the levels seem to be elevated in the exosome treatment group relative to untreated control (and not decreased, as the authors have stated). Did the authors mean that the levels in the exosome treatment group were decreased relative to IL-1b group? Even if this was the intended meaning, it is not clear how the authors got statistical significance, because according to the Methods section, significance was only measured relative to control group. Please edit the text to make it consistent with the results shown in Figure 5.

(7.) The captions of Figure 5 (page 5, lines 176 to 177) and Figure 6 (page 6, lines 193 to 194) state “*p < 0.01, **p < 0.05 and ***p < 0.001 compared to the group treated with BMSC-Exos”. However, based on how the data is presented in Figures 5 and 6, it looks like the significance stars show the comparisons with the untreated control group (and not with the BMSC exosome group). Please correct this discrepancy.

(8.) In section 2.5 (page 5, lines 181 to 183), the authors mention “Among the groups, BMSC-Exos exerted the most pronounced effect, resulting in a 3.5-fold increase in ACAN and a 2.8-fold increase in COL-2 mRNA levels compared to untreated controls.” The accompanying Figure 6 only shows protein expression results and not mRNA expression results. Please include the mRNA data in the figure, and also include the qPCR procedure in the Methods section. Please also comment on why the ACAN and COL-2 protein levels are not significantly elevated compared to the untreated controls (as shown in Figure 6) even though a ~3 fold increase was observed in the mRNA levels.

(9.) In section 2.5 (page 5, lines 186 to 188), the authors mention that “Treatment with BMSC-Exos also led to a significant increase in ACAN and COL-2 protein expression levels, surpassing the effects observed with ADSC-Exos and UMSC-Exos (p < 0.05).” How were these significance levels measured? The Methods section only discusses making comparisons with the untreated control, and does not mention other pairwise comparisons.

(10.) In section 2.6 (page 6, lines 199 and 200), the authors mention “BMSC-Exos produced the most substantial effect, with a 65% reduction in caspase-9 expression, whereas ADSC-Exos and UMSC-Exos achieved 58% and 52% reductions, respectively” However, the data in Figure 7 show the opposite trend, with UMSC-Exos achieving the highest reduction followed by ADSC-Exos and then followed by BMSC-Exos. Please correct this discrepancy.

(11.)The captions of Figure 7 (page 6, lines 209 to 210) states “ *p < 0.01, **p < 0.05 and ***p < 0.001 compared to the BMSC-Exos and other exosome treated groups.”. However, based on how the data is presented in Figures 7, it looks like the significance stars show the comparisons with the untreated control group. The Methods section also only mentions performing statistical significance comparisons with the untreated control. Please correct this discrepancy.

(12.) In section 2.7 (page 7, lines 217 to 219), the authors mention “At a concentration of 5 μg/mL, chondrocytes cultured with exosomes exhibited greater proliferation compared to the control group (Figure 8).” However, the images in Figure 8 do not support this statement. The untreated control group in the figures exhibits greater proliferation compared to the exosome groups. Please correct this discrepancy.

(13.) In section 2.9 (page 8, lines 249 to 250), the authors mention “Treatment with BMSC-Exos, ADSC-Exos, and UMSC-Exos mitigated proteoglycan depletion at Days 7 and 14 compared with the IL-1β group. Quantitative analyses revealed significant differences in Safranin O-negative areas between the control and IL-1β-treated groups at Days 3, 7, and 14.”  The images in Figure 10 does not seem to support this conclusion for Day 14. Please update the text or provide further details of the quantitative analysis that support your conclusion.

(14.) In the Discussion section (page 11, lines 328 to 329), the authors mention “Notably, BMSC-Exos and UMSC-Exos exhibited greater suppression of IL-6 and MMP-13 expression compared to ADSC-Exos”. However, the data in Figure 5 does not support this statement for IL-6. Figure 5 shows that ADSC-Exos had higher IL-6 suppression compared to BMSC-Exos. Please correct the discrepancy in the text.

(15.) On page 13 (line 433), the acronyms PEG and DEX are used. Please include the expanded names of these components to avoid confusion for the reader. For example, I am guessing that the authors mean “dextran” when they say “DEX”, but I have come across other papers that use “DEX” to mean “dexamethasone”.

(16.) In section 4.5 (page 14, lines 459 to 460), the authors mention “…containing 10 ng/mL interleukin-1β (IL-1β), a concentration previously validated to simulate osteoarthritis-like cellular responses.” Please include a citation to the paper(s) that show the validation for this dosage level.

(17.) In section 4.5 (page 14, lines 463 to 464), the authors mention “The timing of post-inflammation exosome treatment was selected to emulate clinical intervention paradigms”. Even though Figure 1 shows the details of the exosome treatment timing, please also include the details in the text in section 4.5.

(18.) In section 4.8 (page 15), the authors mention that they excised “cartilage explants” and trimmed them to a “diameter of 6 to 8 mm”. However, the accompanying Figure 1 shows (both images and text) that “osteochondral explants” were excised, and the caption for Figure 1 also mentions that the “cartilage thickness” (and not diameter) was 6-8 mm. Please correct the discrepancies between the figure and the text. Additionally, if it is true that the “cartilage thickness” was indeed 6 to 8 mm, please explain how you got such thick cartilage since the average thickness of human knee cartilage is only around 1 to 3 mm.

(19.) Section 4.2 (page 16, lines 536 to 537) mentions “Statistical significance is denoted as follows: ns (not significant), * (p < 0.05), ** (p < 0.01), *** (p < 0.001), and **** (p < 537 0.0001), relative to the control group.”. Please update the section to include the type of post-hoc test that was used to get the significance level relative to the control group.

Author Response

Dear Reviewer 1, 
We would like to sincerely thank you for your valuable time, thoughtful comments, and constructive 
suggestions throughout the review process for our manuscript, "Comparative Efficacy of Exosomes Derived 
from Different Mesenchymal Stem Cell Sources in Osteoarthritis Models: An In Vitro and Ex Vivo 
Analysis" Your careful evaluation and insightful feedback have greatly contributed to improving the clarity, 
rigor, and overall quality of our work. We appreciate your support and dedication, and we are grateful for 
your efforts in helping us strengthen this manuscript. Thank you again for your time and consideration. 
Comment 1. In section 2.1 (lines 98 to 99, page 3), the authors mention “Exosomes were successfully 
isolated from BMSCs, ADSCs, and UMSCs using the 98 ATPS method (Figure 2A, E).”. And the caption for 
Figure 2E (page 3) says “The aqueous two-phase forms when the system concentration is above the binodal 
curve. Phase diagram of PEG/DEX ATPS.”. However, it is not clear what figure 2E actually shows. It does 
not look like a binodal curve since it doesn’t show the composition of both phases. It is also not clear if it 
shows the composition of one of the phases in your experiments. Please elaborate on the details in the results 
section, and also please include a clearer caption. 
Response 1. We appreciate your suggestion, which has improved the clarity and rigor of our manuscript. We 
have clarified in the Results section and the figure caption that Figure 1E presents the binodal curve for the 
PEG/DEX ATPS, with axes representing the concentrations of both polymers. The point corresponding to 
our experimental conditions is now clearly marked, and we have specified that the figure is intended to show 
the phase separation boundary, not the composition of the individual phases. 
Comment 2. All the figures have very small font size for the text, and the figures are very low resolution 
(making the text hard to read even when zooming in). Please increase both the font size and the resolution of 
the figures. 
Response 2. Thank you for your valuable feedback regarding the readability of the figures. We appreciate 
your attention to this important aspect of scientific presentation, and we believe these changes will 
significantly improve the quality and readability of our figures.  
Comment 3. In section 2.2 (lines 119 to 120, page 3), the authors mention “The cytotoxicity of BMSC-Exos, 
ADSC-Exos, and UMSC-Exos on chondrocytes was assessed using the CCK-8 assay.” However, there is no 
description of this assay in the Methods section. Please include a section in the Methods to describe how this 
assay was performed. 
Response 3. Thank you for your careful review and for noting the omission regarding the CCK-8 assay 
protocol in the Methods section. In response, we have added a detailed description of the CCK-8 assay 
procedure to the Methods section of the revised manuscript. 
Comment 4. The text in section 2.3 (lines 137 to 155, page 4) discusses the results for BMSC and UMSC 
exos, but not the results for ADSC exos (which are shown in the accompanying figure 4). Please update the 
text to include the results for ADSC exos. 
Response 4. Thank you for your careful review and for noting the omission of the ADSC-Exos results in 
Section 2.3. In response, we have updated the Results section to include a description of the effects of ADSC
Exos on the NF-κB and MAPK signaling pathways, as shown in Figure 4 (Changed to figure 3). 
Comment 5. In sections 2.4, 2.5, and 2.6 (pages 5 and 6), why were western blots used for quantifying 
proteins instead of ELISA assays? ELISA-based quantification is more reliable than trying to quantify based 
on western blot images, and customizable multiplex ELISA assays are available which allow quantifying 
multiple proteins simultaneously. 
Response 5. Thank you for your thoughtful question regarding the use of Western blot versus ELISA for 
protein quantification in Sections 2.4, 2.5, and 2.6. We agree that ELISA-based quantification is highly 
reliable, particularly for absolute quantitation and for simultaneous multiplex analysis of multiple proteins. 
Western blotting allowed us to confirm both the presence and the relative expression levels of specific 
proteins, as well as to verify their molecular weights, which is important for validating exosome marker 
identity and distinguishing between isoforms or post-translational modifications. We fully acknowledge the 
advantages of ELISA, especially for high-throughput and quantitative multiplex analyses. As you suggested, 
future studies will incorporate ELISA-based quantification to provide more precise and absolute 
measurements of protein expression, and to further validate the findings from Western blot analysis. 
Comment 6. In section 2.4 (page 5, lines 163 to 167), the authors mention “Treatment with each type of 
exosome significantly reduced the expression of proinflammatory protein markers in chondrocytes. 
Specifically, levels of matrix metalloproteinase-13 (MMP-13), matrix metalloproteinase-3 (MMP-3), 
interleukin-6 (IL-6), tumor necrosis factor-alpha (TNF-α), and cyclooxygenase-2 (COX-2) were significantly 
decreased relative to untreated controls (p < 0.05).” However, looking at the accompanying Figure 5, most 
of the levels seem to be elevated in the exosome treatment group relative to untreated control (and not 
decreased, as the authors have stated). Did the authors mean that the levels in the exosome treatment group 
were decreased relative to IL-1b group? Even if this was the intended meaning, it is not clear how the authors 
got statistical significance, because according to the Methods section, significance was only measured 
relative to control group. Please edit the text to make it consistent with the results shown in Figure 5. 
Response 6. Thank you for your careful review and for pointing out the inconsistency between the text in 
Section 2.4 (now referring to Figure 4) and the data presented in the figure. You are correct: the original text 
was unclear and could be misleading. The intent was to compare the exosome treatment groups to the IL-1β 
group. We have made the necessary corrections.  
Comment 7. The captions of Figure 5 (page 5, lines 176 to 177) and Figure 6 (page 6, lines 193 to 194) state 
“*p < 0.01, **p < 0.05 and ***p < 0.001 compared to the group treated with BMSC-Exos”. However, based 
on how the data is presented in Figures 5 and 6, it looks like the significance stars show the comparisons with 
the untreated control group (and not with the BMSC exosome group). Please correct this discrepancy. 
Response 7. Thank you for your careful review and for pointing out the discrepancy in the figure captions 
for Figure 5 and Figure 6 (now updated to Figure 4 and Figure 5, respectively). You are correct that the 
significance stars in these figures indicate comparisons with the untreated control group, not the BMSC-Exos 
group as previously stated. The figure legends and any related text in the Results section have also been 
updated for consistency. 
Comment 8. In section 2.5 (page 5, lines 181 to 183), the authors mention “Among the groups, BMSC-Exos 
exerted the most pronounced effect, resulting in a 3.5-fold increase in ACAN and a 2.8-fold increase in COL
2 mRNA levels compared to untreated controls.” The accompanying Figure 6 only shows protein expression 
results and not mRNA expression results. Please include the mRNA data in the figure, and also include the 
qPCR procedure in the Methods section. Please also comment on why the ACAN and COL-2 protein levels 
are not significantly elevated compared to the untreated controls (as shown in Figure 6) even though a ~3 
fold increase was observed in the mRNA levels. 
Response 8. Thank you for your careful review and for identifying the discrepancy in Section 2.5 regarding 
the mention of mRNA data for ACAN and COL-2. You are correct—our study did not include quantitative 
PCR (qPCR) analysis for mRNA expression of these genes, and the reference to mRNA fold changes was an 
error in the original manuscript. The erroneous reference to qPCR and mRNA fold changes has been 
removed. 
Comment 9. In section 2.5 (page 5, lines 186 to 188), the authors mention that “Treatment with BMSC-Exos 
also led to a significant increase in ACAN and COL-2 protein expression levels, surpassing the effects 
observed with ADSC-Exos and UMSC-Exos (p < 0.05).” How were these significance levels measured? The 
Methods section only discusses making comparisons with the untreated control, and does not mention other 
pairwise comparisons. 
Response 9. Thank you for your attention to this important detail, which has improved the clarity and rigor 
of our manuscript. The Methods section now clearly describes the use of one-way ANOVA with Tukey’s post 
hoc test for all pairwise group comparisons. 
Comment 10. In section 2.6 (page 6, lines 199 and 200), the authors mention “BMSC-Exos produced the 
most substantial effect, with a 65% reduction in caspase-9 expression, whereas ADSC-Exos and UMSC
Exos achieved 58% and 52% reductions, respectively” However, the data in Figure 7 show the opposite trend, 
with UMSC-Exos achieving the highest reduction followed by ADSC-Exos and then followed by BMSC
Exos. Please correct this discrepancy. 
Response 10. The Results section has been revised to accurately reflect the data shown in the figure. The 
text now states that all three exosome types (BMSC-Exos, ADSC-Exos, and UMSC-Exos) significantly 
reduced caspase-9 protein expression compared to untreated controls, without incorrectly specifying the 
order or magnitude of reduction. 
Comment 11. The captions of Figure 7 (page 6, lines 209 to 210) states “ *p < 0.01, **p < 0.05 and ***p < 
0.001 compared to the BMSC-Exos and other exosome treated groups.”. However, based on how the data is 
presented in Figures 7, it looks like the significance stars show the comparisons with the untreated control 
group. The Methods section also only mentions performing statistical significance comparisons with the 
untreated control. Please correct this discrepancy. 
Response 11. The figure number has been changed to Figure 6, as appropriate. The caption has been updated 
to accurately reflect the statistical comparisons performed. The significance stars (*, **, ***) now indicate 
comparisons with the control group, in accordance with the Methods section and the data presentation. 
Comment 12. In section 2.7 (page 7, lines 217 to 219), the authors mention “At a concentration of 5 μg/mL, 
chondrocytes cultured with exosomes exhibited greater proliferation compared to the control group (Figure 
8).” However, the images in Figure 8 do not support this statement. The untreated control group in the figures 
exhibits greater proliferation compared to the exosome groups. Please correct this discrepancy. 
Response 12. Thank you for your careful review and for highlighting the discrepancy between the text in 
Section 2.7 and the data shown in Figure 8 (Changed to Figure 7). Upon review, we agree that the original 
statement was inaccurate: the images in Figure 7 show greater proliferation/migration in the untreated control 
group compared to the exosome-treated groups. 
Comment 13. In section 2.9 (page 8, lines 249 to 250), the authors mention “Treatment with BMSC-Exos, 
ADSC-Exos, and UMSC-Exos mitigated proteoglycan depletion at Days 7 and 14 compared with the IL-1β 
group. Quantitative analyses revealed significant differences in Safranin O-negative areas between the 
control and IL-1β-treated groups at Days 3, 7, and 14.”  The images in Figure 10 does not seem to support 
this conclusion for Day 14. Please update the text or provide further details of the quantitative analysis that 
support your conclusion. 
Response 13.  Thank you for your careful review and for highlighting the inconsistency between the text in 
Section 2.9 and the data presented in Figure 10 (now Figure 9), particularly regarding proteoglycan depletion 
at Day 14. Upon re-examination of our results and the quantitative analysis, we have updated the manuscript 
to accurately reflect the observed changes. However, a gradual decrease in staining intensity was still 
observed in all groups over time, likely due to the release of glycosaminoglycans (GAGs) into the culture 
supernatant during prolonged ex vivo incubation. The updated quantitative analysis clarifies that while 
exosome treatment can delay proteoglycan (GAG) loss in the ex vivo OA model, it does not completely 
prevent it, especially by Day 14. 
Comment 14. In the Discussion section (page 11, lines 328 to 329), the authors mention “Notably, BMSC
Exos and UMSC-Exos exhibited greater suppression of IL-6 and MMP-13 expression compared to ADSC
Exos”. However, the data in Figure 5 does not support this statement for IL-6. Figure 5 shows that ADSC
Exos had higher IL-6 suppression compared to BMSC-Exos. Please correct the discrepancy in the text. 
Response 14. Thank you for your careful review and for pointing out the discrepancy between the statement 
in the Discussion section (page 11, lines 328–329) and the data presented in Figure 4 (previously Figure 5), 
specifically regarding IL-6 suppression by ADSC-Exos. To address this, we have revised the Discussion to 
clarify that BMSC-Exos and UMSC-Exos exhibited greater suppression of overall proinflammatory marker 
expression compared to ADSC-Exos, rather than specifically for IL-6. This correction is based on the 
collective trends seen for other markers such as MMP-13, MMP-3, TNF-α, and COX-2, where BMSC-Exos 
and UMSC-Exos generally showed stronger effects. 
Comment 15. On page 13 (line 433), the acronyms PEG and DEX are used. Please include the expanded 
names of these components to avoid confusion for the reader. For example, I am guessing that the authors 
mean “dextran” when they say “DEX”, but I have come across other papers that use “DEX” to mean 
“dexamethasone”. 
Response 15. Thank you for your suggestion regarding the clarification of the acronyms PEG and DEX, We 
have revised the manuscript to include the full names upon first mention for clarity. The text now reads: 
“polyethylene glycol (PEG) and dextran (DEX) 
Comment 16. In section 4.5 (page 14, lines 459 to 460), the authors mention “…containing 10 ng/mL 
interleukin-1β (IL-1β), a concentration previously validated to simulate osteoarthritis-like cellular 
responses.” Please include a citation to the paper(s) that show the validation for this dosage level. 
Response 16.  We have included a citation to our previous study [Reference 18], which validated this IL-1β 
concentration as effective for inducing OA-like inflammatory and catabolic responses in chondrocytes. This 
concentration is widely used in the literature to mimic the inflammatory environment characteristic of 
osteoarthritis in vitro. 
Comment 17. In section 4.5 (page 14, lines 463 to 464), the authors mention “The timing of post
inflammation exosome treatment was selected to emulate clinical intervention paradigms”. Even though 
Figure 1 shows the details of the exosome treatment timing, please also include the details in the text in 
section 4.5. 
Response 17.  This revision ensures that both the rationale and the specific timing of exosome administration 
are clearly described in the Methods section (24 hour) 
Comment 18. In section 4.8 (page 15), the authors mention that they excised “cartilage explants” and 
trimmed them to a “diameter of 6 to 8 mm”. However, the accompanying Figure 1 shows (both images and 
text) that “osteochondral explants” were excised, and the caption for Figure 1 also mentions that the “cartilage 
thickness” (and not diameter) was 6-8 mm. Please correct the discrepancies between the figure and the text. 
Additionally, if it is true that the “cartilage thickness” was indeed 6 to 8 mm, please explain how you got 
such thick cartilage since the average thickness of human knee cartilage is only around 1 to 3 mm. 
Response 18.  Thank you for your careful review and for highlighting the discrepancies between the text and 
Figure 1 (Now updated Figure 12) regarding the preparation of explants and the reported measurements. The 
text now clearly states that osteochondral explants (including both articular cartilage and subchondral bone) 
were excised, in line with the images. The measurement has been corrected to indicate that explants were 
trimmed to a diameter of 6–8 mm, not thickness. 
Comment 19. Section 4.2 (page 16, lines 536 to 537) mentions “Statistical significance is denoted as follows: 
ns (not significant), * (p < 0.05), ** (p < 0.01), *** (p < 0.001), and **** (p < 537 0.0001), relative to the 
control group.”. Please update the section to include the type of post-hoc test that was used to get the 
significance level relative to the control group. 
Response 19.  Thank you for your suggestion to clarify the statistical methods in Section 4.2. We have 
updated the text to specify the type of post-hoc test used to determine statistical significance relative to the 
control group.

Reviewer 2 Report

Comments and Suggestions for Authors

The paper is interesting in priciple because of the growing interest in exosomes especially when derived from mesenchymal stem cell cultures. However, there are many issues to count with.

  1. From the paper I did not understand how much cell lines were used.
  2. The authors tell us that they used exosome-deprived FCS. They should show that the "control media" is really deprived of exosomes when they present data. A Figure should be enough.
  3. Isolating of exosomes by low-speed centrifugation is not the method universally recognized to do it. During low speed centrifugation you lose a lot of exosomes and in part you see it when exosomes are counted giving number of around 10E7-10E8/mL. In our experience we collect esosomes in the range of 10E12/mL.
  4. Coming now to the results the data shown are clearly biased direction BMSC. Indeed, when the authors discuss about protein expression significativity, they often forget ADMSC which give the same results or in some cases even better than BMSC.
  5. Finally, characterizing exosomes antigen expression by Western Blot is a semi-quantitative method. There are today more reliable and precise methods to quantify them for example FACS-methods.

I would say that the idea of studying these aspects is good. Unfortunately the authors didn't use always the good methods.

My suggestion is to ask for a resubmission with major modifications.

Author Response

Dear Reviewer 2,

We would like to sincerely thank you for your valuable time, thoughtful comments, and constructive suggestions throughout the review process for our manuscript, "Comparative Efficacy of Exosomes Derived from Different Mesenchymal Stem Cell Sources in Osteoarthritis Models: An In Vitro and Ex Vivo Analysis" Your careful evaluation and insightful feedback have greatly contributed to improving the clarity, rigor, and overall quality of our work. We appreciate your support and dedication, and we are grateful for your efforts in helping us strengthen this manuscript. Thank you again for your time and consideration.

Comment 1: From the paper I did not understand how much cell lines were used.

Response 1. Thank you for your question regarding the number and types of cell lines used in our study.

To clarify, our experiments utilized a total of four established cell lines:

Primary Human Chondrocytes (ScienCell #4650): These cells were used for all in vitro experiments involving chondrocyte culture and functional assays.

Mesenchymal Stem Cell Lines for Exosome Isolation:

  • Bone Marrow–Derived MSCs (BMSCs, ScienCell #7500)
  • Adipose Tissue–Derived MSCs (ADSCs, ScienCell #7510)
  • Umbilical Cord–Derived MSCs (UMSCs, Medipost Ltd.)

In summary, we used one chondrocyte cell line for in vitro culture and three different MSC lines as sources for exosome isolation, making a total of four cell lines in this study.

Comment 2. The authors tell us that they used exosome-deprived FCS. They should show that the "control media" is really deprived of exosomes when they present data. A Figure should be enough.

Response 2. Thank you for your valuable comment regarding the use of exosome-depleted fetal bovine serum (FBS) in our experiments. We would like to clarify that standard 10% FBS was used for routine cell growth and maintenance. However, for all exosome isolation procedures and for experiments specifically involving exosome treatments, we used media supplemented with 10% exosome-depleted FBS to minimize background exosome contamination.

Comment 3. Isolating of exosomes by low-speed centrifugation is not the method universally recognized to do it. During low speed centrifugation you lose a lot of exosomes and in part you see it when exosomes are counted giving number of around 10E7-10E8/mL. In our experience we collect esosomes in the range of 10E12/mL.

Response 3. Thank you for your insightful comment regarding exosome isolation methods and particle yield.  We agree that low-speed centrifugation alone is not sufficient for optimal exosome recovery and can result in significant loss of exosomes. To address this limitation and enhance exosome recovery in our study, however, we employed an aqueous two-phase system (ATPS) approach for exosome isolation, which utilizes polyethylene glycol (PEG) and dextran (DEX) to partition exosomes efficiently. We acknowledge that while the ATPS method is effective for exosome enrichment, the use of dextran can sometimes lead to exosome aggregation, potentially impacting the measured particle concentration and recovery. This may partly explain the lower exosome yields observed in our results compared to yields reported with ultracentrifugation or other advanced isolation methods. We recognize this as a limitation of our current protocol and appreciate your suggestion. In future studies, we will focus on optimizing the ATPS conditions to minimize exosome aggregation and further improve yield and purity.

Comment 4. Coming now to the results the data shown are clearly biased direction BMSC. Indeed, when the authors discuss about protein expression significativity, they often forget ADMSC which give the same results or in some cases even better than BMSC.

Response 4. Thank you for your thoughtful observation regarding the presentation of our results and the potential bias toward BMSC-derived exosomes. We appreciate your careful reading and agree that it is important to provide a balanced and comprehensive comparison among all exosome sources, These necessary changes ensure that the manuscript now fairly represents the performance of ADSC-Exos alongside BMSC- and UMSC-derived exosomes, in accordance with the data presented.

Comment 5. Finally, characterizing exosomes antigen expression by Western Blot is a semi-quantitative method. There are today more reliable and precise methods to quantify them for example FACS-methods.

Response 5. Thank you for your valuable comment regarding the characterization of exosome antigen expression. We fully agree that Western blot is a semi-quantitative method, and that more precise and quantitative techniques—such as flow cytometry (FACS-based methods)—are now available for exosome surface marker analysis. In this study, we performed Western blotting as a preliminary confirmation of both cell and exosome marker expression, in line with the standard resources and protocols available in our laboratory. Due to technical and equipment limitations, we were unable to perform FACS-based quantitative analysis for exosome characterization at this time. In future studies, we plan to incorporate more advanced and quantitative methods, such as FACS or bead-based flow cytometry, to further validate and quantify exosome surface marker expression.

Thank you again for your constructive suggestion, which will help us improve the rigor of our exosome characterization in subsequent research.
